# CC5 and CC8, Two Disintegrin Isoforms from *Cerastes cerastes* Snake Venom Decreased Inflammation Response In Vitro and In Vivo

**DOI:** 10.3390/ijms241512427

**Published:** 2023-08-04

**Authors:** Maram Morjen, Ons Zakraoui, Zaineb Abdelkafi-Koubaa, Najet Srairi-Abid, Naziha Marrakchi, Khadija Essafi-Benkhadir, Jed Jebali

**Affiliations:** 1Laboratory of Biomolecules, Venoms and Theranostic Applications, LR20IPT01, Pasteur Institute of Tunis, University of Tunis El Manar, Tunis 1002, Tunisia; abdelkafi_zaineb@yahoo.fr (Z.A.-K.); najet.abidsrairi@pasteur.tn (N.S.-A.); naziha.marrakchi@pasteur.tn (N.M.); 2Laboratory of Molecular Epidemiology and Experimental Pathology, LR16IPT04, Pasteur Institute of Tunis, University of Tunis El Manar, Tunis 1002, Tunisia; zakraoui-ons@hotmail.fr (O.Z.); khadija.essafi@pasteur.tn (K.E.-B.); 3Research Laboratory of Precision Medicine/Personalized Medicine and Oncology Investigation, LR21SP01, Salah Azaiez Institute, University of Tunis El Manar, Tunis 1007, Tunisia; 4Medicine School of Tunis, University of Tunis El Manar, 15 Djebel Lakhdhar Street, La Rabta, Tunis 1007, Tunisia

**Keywords:** snake venom, disintegrin, inflammation, cytokines, signaling pathways

## Abstract

Inflammation is associated with many pathology disorders and the malignant progression of most cancers. Therefore, targeting inflammatory pathways could provide a promising strategy for disease prevention and treatment. In this study, we experimentally investigated the anti-inflammatory effect of CC5 and CC8, two disintegrin isoforms isolated from *Cerastes cerastes* snake venom, on LPS-stimulated macrophages, both on human THP-1 and mouse RAW264.7 cell adherence and their underlying mechanisms by measuring cytokine release levels and Western blot assay. Equally, both molecules were evaluated on a carrageenan-induced edema rat model. Our findings suggest that CC5 and CC8 were able to reduce adhesion of LPS-stimulated macrophages both on human THP-1 and mouse RAW264.7 cells to fibrinogen and vitronectin through the interaction with the αvβ3 integrin receptor. Moreover, CC5 and CC8 reduced the levels of reactive oxygen species (ROS) mediated by the NF-κB, MAPK and AKT signaling pathways that lead to decreased production of the pro-inflammatory cytokines TNF-α, IL-6 and IL-8 and increased secretion of IL-10 in LPS-stimulated THP-1 and RAW264.7 cells. Interestingly, both molecules potently exhibited an anti-inflammatory effect in vivo by reducing paw swelling in rats. In light of these results, we can propose the CC5 and CC8 disintegrins as interesting tools to design potential candidates against inflammatory-related diseases.

## 1. Introduction

Inflammation underlies the immune response to biological, chemical, or physical stimuli for the healing process recovery under stressful conditions [1]. Nevertheless, persistent inflammation leads to a chronic stage and may initiate numerous diseases, such as cancer, asthma, diabetes, atherosclerosis, as well as autoimmune, cardiovascular, neurodegenerative and metabolic diseases [2]. The inflammatory process is initiated by the interactions between various effectors, associated with activation of signaling pathways that modulate expression of both pro- and anti-inflammatory mediators [3,4]. Extensive research demonstrated that nuclear factor-κB (NF-κB) is a significant mediator of inflammation as it regulates large arrays of genes encoding cytokines, cytokine receptors, and cell adhesion molecules that are involved in triggering inflammation [5,6]. Among cell adhesion molecules, integrins are involved in multiple stages of immune and inflammatory responses [7]. They mainly mediate adhesive interactions, coordinate the ingestion of extracellular matrix components, and regulate different selective cell responses, such as cytoskeleton formation, transmigration into the inflammatory site, cytokine secretion, and production or reactive oxygen intermediates, thereby regulating the mitogen-activated protein kinase (MAPK) and PI3K/Akt signaling pathways in neutrophils and macrophages [8,9].

Anti-integrin therapies, with gut selectivity, offer a new class of therapeutics that are safe, well tolerated and hold significant promise for efficacy in inflammatory related diseases. Indeed, clinical data showed that targeted anti-integrin therapies, such as vedolizumab, abrilumab, etrolizumab, offer a promising alternative for the treatment of inflammatory bowel disease (IBD) [10,11]. The elucidation of pivotal triggers in inflammatory-related disease like integrins, tumor necrosis factor alpha (TNFα), as well as specific cytokines like interleukin (IL)-6, in inflammatory-related diseases launched the successful development of new pharmacological approaches, leading to an improvement in therapeutic outcomes. Interestingly, snake venom comprises a variety of protein families, such as the C-type lectins [12], Kunitz-type serine protease inhibitor [13,14], and disintegrins [15], that bind and interfere with integrins. The disintegrins family comprises a group of cysteine-rich proteins (40–100 amino acids). The “classical” disintegrins are either monomeric or dimeric and possess an XGD motif, with X being Arg, Lys, Met, Trp, or Val residues. Other motifs have been noticed, such as MLD, MVD, or K/RTS sequences, in their active site motifs [16]. These tripeptide sequences, known as the “RGD adhesive loop”, possess a critical role in interactions with integrin receptors that play a fundamental role in mediating physiological and pathological processes, such as hemostasis and cancer [17,18,19]. Thus, disintegrins are capable of inhibiting the aggregation of human platelets as well as tumor growth and progression in addition to angiogenesis, apoptosis, and inflammation. Thereby, disintegrins are reviewed as antithrombotic and antitumoral agents [20].

CC5 and CC8 are two highly homologous dimeric disintegrins isolated from the Tunisian *Cerastes cerastes* viper venom. As reported by Calvete et al., CC5 is monomeric and contains an RGD motif in its two subunits, whereas CC8 is heterodimeric and possess an RGD and a WGD sequences in its α and β subunits, respectively. Interestingly, CC5 and CC8 were able to inhibit platelet aggregation and cell adherence by interfering αIIbβ3, α_v_β_3_ and α5β1 integrins [15]. Furthermore, our team reported that CC5 and CC8 exhibited a significant anti-angiogenic effect both in vitro and ex vivo and displayed pro-apoptotic activity in endothelial HMEC cells by inducing caspase-3 activation and FAK/PI3K/AKT down-regulation [21]. In this study, we investigate, for the first time, the anti-inflammatory effect of CC5 and CC8 in LPS-stimulated RAW264.7 and human-derived macrophage THP-1 cells in vitro as well as their underlying mechanisms. The anti-inflammatory effect was also evaluated in vivo by a carrageenan-induced edema in rat model.

## 2. Results

### 2.1. CC5 and CC8 Effect on THP-1-Derived Macrophages and RAW264.7 Cells Development

To investigate the role of CC5 and CC8 on the regulation of inflammatory response, we first evaluated their effect on the viability of RAW264.7 and THP-1-derived macrophages to identify doses that did not generate any effect on cell viability. We used the MTT assay to quantify living cells and demonstrate that CC5 and CC8 did not affect the viability of RAW264.7 and differentiated THP-1 macrophages, up to 24 h of treatment, until a concentration of 200 nM (Figure 1). This result was also confirmed by Trypan blue assay (Appendix A).

Because integrin-mediated adhesion to the extracellular matrix (ECM) is essential during macrophage recruitment to the site of inflammation, we investigate the effect of CC5 and CC8 on THP-1 and RAW264.7 cell adhesion to various extracellular matrices (ECM). As shown in (Figure 2A(a),B(a), CC5 and CC8 did not affect the adherence of non-stimulated THP-1 and RAW264.7 cells seeded on various ECM, except on vitronectin (Vn), and the data show in Figure 2A(a) that CC5 and CC8 exhibited a significant slight amplification of THP-1 cell adherence, about 13.4% and 19.9%, respectively. Yet, in Figure 2B(a), only CC8 induce a significant slight diminution on RAW cell adhesion of about 13.9%. However, both CC5 and CC8 decreased the adhesion of LPS-stimulated RAW264.7 and THP-1 macrophages to fibrinogen (Fg) and vitronectin (Vn), whereas no effect was observed on Collagen I. A significant slight decrease in the adhesion of cells to poly-L-lysine was noticed for CC5 at concentrations of 25, 50, and 100 nM and CC8 at 100 nM about 14.3%, 22.6%, 6%, and 9.4%, respectively, for THP-1 cells and for CC5 and CC8 at concentrations of 25, 50, and 100 nM at about 31%, 22%, 29%, 11.7%, 25.6%, and 20%, respectively, for RAW264.7. The CC5 maximal inhibitory effect on Fg and Vn was about 53.6% and 37.1%, respectively, for THP-1 cells and 61.6% and 64.6%, respectively, for RAW264.7 cells. For CC8, inhibition values were about 66% and 53.8%, respectively, for THP-1 cells and 64.7% and 70.6%, respectively, for RAW264.7 cells at a dose of 100 nM (Figure 2A(b,c),B(b,c)). This finding suggests that CC5 and CC8 could affect cell adhesion through αvβ3 integrin cell receptor reported to be the binding site for disintegrins in activated macrophages [22,23].

As shown in Figure 3, in Western blot assays using monoclonal antibodies against αv and β3 integrins, we found that CC5 and CC8 reduced the protein levels of the αvβ3 integrin subunits in both LPS-stimulated RAW264.7 and THP-1cells. This result supports the implication of αvβ3 integrin in the inhibitory effect of CC5 and CC8 on LPS-stimulated THP-1 and RAW264.7 cell adherence towards fibrinogen and vitronectin.

### 2.2. CC5 and CC8 Inhibited LPS-Induced Cytokine Production in THP-1 and RAW264.7 Cells

Cytokine release from macrophages is considered as an inflammatory response to stimulus. Interleukin (IL)-6, tumor necrosis factor (TNF), and IL-8 are well known pro-inflammatory cytokines which are involved in local and systemic inflammatory reactions, while IL-10, as an anti-inflammatory cytokine, is reported to block the production of the monokines (IL-1, IL-6, and TNF-α) in response to lipopolysaccharide or other stimuli [24]. Thus, THP-1 and RAW264.7 macrophages were firstly stimulated by LPS (1 μg/mL) to investigate whether CC5 and CC8 decreased the expression of inflammatory cytokines, such as IL-6, TNFα, IL-8, and IL-10 after 24 h. As shown in Figure 4A,B in black bars, the levels of inflammatory cytokines is significantly increased in LPS-stimulated THP-1 and RAW264.7 cells. In contrast, macrophage treatment with CC5 and CC8 (25 and 50 nM) reduced the release of IL-6, TNFα, and IL-8 cytokines (Figure 4A(a–c),B(a–c) and enhanced the production of IL-10 in a dose-dependent manner (Figure 4A(d),B(d)).

### 2.3. Modulation of LPS-Induced Inflammatory Cytokine Production by RAW264.7 and THP-1 Cells Is Mediated by NF-κB, ERK1/2, P38 MAPK, and Akt Inhibition

Increasing evidence suggests that LPS activates the mitogen-activated protein kinase (MAPK) signaling pathway and the transcription factor nuclear factor κB (NF-κB) in macrophages [25], leading to the release of inflammatory cytokines, including tumor necrosis factor-α (TNF-α) and interleukins (IL)-1, IL-6, IL-8, and IL-12 [26]. In this context, we evaluated the effect of CC5 and CC8 on LPS-stimulated activation of NF-κB, AKT, P38, and ERK1/2 in THP-1 and RAW264.7 macrophages using Western blot assay. As shown in Figure 5, immunoblot analysis indicated that stimulation of cells with LPS (1 µg/mL) during 24 h, increased the phosphorylation levels of NF-κB, Akt, ERK1/2, and p38 compared to unstimulated cells. Interestingly, a significant reduction in the phosphorylated forms of these effectors was observed on LPS-stimulated macrophages treated with CC5 and CC8 (25 nM and 50 nM) (Figure 5). Our results suggest that CC5 and CC8 interact with αvβ3 integrin receptor on LPS-stimulated macrophages, thereby modulating the NF-κB, MAPK, and Akt pathways to regulate inflammatory response.

### 2.4. CC5 and CC8 Reduced ROS Production in LPS-Stimulated RAW264.7 and THP-1 Cells

To investigate the impact of CC5 and CC8 on LPS-induced ROS in RAW 264.7 and THP-1-derived macrophages, we measured their intracellular levels using fluorescent cellular reactive oxygen species detection assay. As demonstrated in Figure 6, ROS production was increased in LPS-stimulated cells and in positive control cells treated with 1 mM of H_2_O_2_. In contrast, CC5 and CC8 (at 50 nM) decreased LPS-induced intracellular ROS generation in RAW cells to about 19.3% and 20.8%, respectively, and to 17.6% and 21%, respectively, in THP-1 cells. This finding reveals that CC5 and CC8 reduce oxidative stress in LPS- stimulated macrophages and modulate inflammatory signaling pathway regulated by NF-κB, thereby inhibiting the expression of pro-inflammatory cytokines, such as TNF-α and IL-6 and IL-8.

### 2.5. Effects of CC5 and CC8 on Carrageenan-Induced Paw Edema in Rats

In order to evaluate the anti-inflammatory effect of CC5 and CC8 in vivo, we used a carrageenan-induced paw edema model in rats which is one of the well-established acute inflammatory models in vivo. After injecting carrageenan into the rat’s hind paw, an inflammatory response was highlighted by an increase in paw edema which persisted for at least 4 h. Dexamethasone, used as positive control, was able to reduce the inflammatory response induced by carrageenan injection in rats. Interestingly, CC5 and CC8 reduced paw swelling from the second hour in disintegrins-treated-rats. This reduction was about 24.9% for CC5 and 28.8% for CC8 after the second hour and reached a maximum of 34.3% for CC5 and 37% for CC8 at the fourth hour (Figure 7). According to the carrageenan-induced paw edema in rat model, we have established the potential effect of CC5 and CC8 in reducing inflammatory response in vivo by decreasing paw swelling.

## 3. Discussion

Many studies reported the main role of integrins in inflammation through allowing cell adhesion, regulating communication with the microenvironment, and modulating cell signaling [27,28]. In our study, we demonstrated that both snake venom disintegrins CC5 and CC8 affected the expression αvβ3 integrin subunits and were able to reduce RAW264.7 cells and THP-1-derived macrophages adhesion on fibrinogen and vitronectin, thereby regulating inflammatory response. Nevertheless, the significant slight decrease of cell adherence on poly-L-lysine (an integrin independent substratum) suggests that other cellular effectors may be involved in the biological activity of CC5 and CC8 on RAW264.7 cells and THP-1-derived macrophages. Thus, our results are in accordance with the studies reporting the implication of integrins including αvβ3 on inflammation related diseases [9,29]. Indeed, αvβ3 integrin was reported to be the most prominent receptor affecting tumor growth, tumor invasiveness, metastasis, angiogenesis, inflammation, osteoporosis, and rheumatoid arthritis [30,31].

Interestingly, the inhibitory effect of CC5 and CC8 at 100 nM equivalent to (≃1.4 µg/mL) on the adherence of LPS-stimulated RAW264.7 and THP-1 macrophages was obtained with concentration seven fold lower than used with trimucrin (10 µg/mL), a disintegrin from the venom of *Trimeresurus mucrosquamatus.* This activity was mediated by the αvβ3 integrin [22]. CC8 is also 20-fold more active than rhodostomin (30 µg/mL), a disintegrin from *Calloselasma rhodostoma* snake venom, which inhibited cell adhesion and migration and interacts with the αvβ3 integrin receptor of monocytes/macrophages [23]. These results are in favor that the disintegrins CC5 and CC8 may provide a strategy for the development of treatment against inflammatory-related diseases by targeting αvβ3 integrin.

Macrophages constitute the first line of defense as response to a stimulus and rapidly produce large amounts of inflammatory cytokines [32]. The production of these cytokines can initiate a cascade of inflammatory mediator release, and often several cytokines are required to synergize to exhibit an optimal function leading to determine the nature of immune responses and to control immune cell trafficking and the cellular arrangement of immune organs [33]. In fact, pro-inflammatory cytokines, such as interleukins (IL-6, IL-8) and tumor necrosis factor-α (TNF-α), initiate a profound immune response and are involved in the up-regulation of inflammatory reactions, whereas anti-inflammatory cytokines regulate the immune response by controlling pro-inflammatory cytokine responses [34]. Among all the anti-inflammatory cytokines, IL-10 plays a pivotal role in repressing the expression of inflammatory cytokines, including IL-6 and TNF-α produced by activated macrophages [35]. Down-regulation of pro-inflammatory cytokine receptors is the hallmark of IL-10 expression. Thus, this cytokine is able to regulate the release and function of pro-inflammatory molecules at multiple levels [36]. Interestingly, our results showed that CC5 and CC8 modulate negatively the pro-inflammatory cytokines IL-6, TNF-α, and IL-8 levels through increasing the production of IL-10 in both THP-1-derived macrophages and RAW264.7 cells. The net outcome of this cross-talk between the “pro-” and “anti-inflammatory” arms of the immune response is the down-regulation of the excessive and harmful inflammation in infections, preventing host-mediated tissue destruction and controlling the resolution of inflammation [34]. Previous studies focusing on anti-inflammatory agents purified from snake venom demonstrated that trimucrin decreased the release of the pro-inflammatory cytokines TNFα and IL-6 in LPS-activated macrophages [22]. Furthermore, rhodostomin decreased the production of TNF-α, (IL)-6, IL-1β, and IL-10 in LPS-induced endotoxemic mice [23]. From *Macrovipera lebetina* snake venom, our team has previously reported that a C-type lectin protein, lebecetin, reduced the LPS-induced pro-inflammatory cytokine production in human THP-1-derived macrophages [37].

It is well known that activation of the MAPK and NF-κB signaling pathways in stimulated macrophages leads to the release of inflammatory cytokines [25,26]. In this work, we have highlighted the role of CC5 and CC8 in decreasing the phosphorylation levels of NF-κB, ERK1/2, P38, and AKT kinases. Thus, our results suggest that CC5 and CC8 could interact with αvβ3 integrin receptor on LPS-stimulated macrophages, thereby modulating the NF-κB, MAPK, and Akt pathways to regulate inflammatory response and reducing cytokine release. This finding is in agreement with results reported by Hung et al. showing that trimucrin inhibited the activation of the NF-κB and MAPK pathways through interaction with αvβ3 integrin in murine RAW264.7 and human THP-1 macrophages [22]. Moreover, other studies have indicated that rhodostom interacted with the αvβ3 integrin of monocytes/macrophages, leading to interference with the LPS-induced activation of the NFκB and MAPK pathways [23]. Therefore, targeting pro-inflammatory signaling pathways may be useful to identify novel therapeutic strategies for chronic disease treatment. 

Previous studies reported that oxidative stress, caused by the production of reactive oxygen species (ROS), plays a pivotal role in inflammatory processes [38,39]. Oxidative stress can activate a variety of transcription factors leading to the expression of some genes involved in inflammatory pathways, especially the nuclear factor kappa B (NF-κB). ROS drive modifications of IκB proteins, leading to their degradation and allowing active NF-κB translocation to the nucleus and inducing inflammatory cytokine release, such as (TNF)-α, IL-1β, and IL-6, which are involved in the inflammatory process of acute lung injury (ALI) [40,41]. Our results suggest that CC5 and CC8 reduce ROS production in LPS-stimulated macrophages through modulating inflammatory signaling pathway regulated by NF-κB and thereby inhibiting the expression of pro-inflammatory cytokines such as TNF-α and IL-6 and IL-8. 

Considering the promising results obtained in vitro, we have investigated the anti-inflammatory effect of CC5 and CC8 in vivo. Using the acute inflammatory model in rats, we found that CC5 and CC8 reduced the formation of edema induced by carrageenan. Moreover, we observed a significant decrease in paw swelling upon the second hour. Further studies are needed to consolidate our findings, including biomarker estimation using ELISA and RT-qPCR techniques to measure levels of various pro- and anti-inflammatory cytokines such as (TNF alpha, IL6….) in serum samples. Also, histopathological studies from different paw samples are necessary to validate potential role of CC5 and CC8 on reduction in pannus formation, joint swelling, and synovial hyperplasia. On the other side, the study of Hsu et al. [23] reported that the protective function of the snake venom protein rhodostomin against tissue inflammation in LPS-induced endotoxemia may be attributed to its anti-inflammatory activity in vivo. 

## 4. Materials and Methods

### 4.1. Cell Culture

RAW264.7 (mouse macrophages) and THP-1 (human monocytes) cells were obtained from the American Type Culture Collection (Manassas, VA, USA). RAW264.7 cells were maintained in RPMI-1640 media supplemented with 10% fetal bovine serum (Gibco, Billings, MT, USA), and THP-1 cells, were cultured in RPMI 1640/ Glutamax-1 medium (Invitrogen Life Technologies, Carlsbad, CA, USA) with 10% fetal bovine serum, 2% non-essential amino acids (Sigma, St. Louis, MO, USA), 100 U/mL penicillin, 1% sodium pyruvate (Gibco), 100 µg/mL streptomycin (Gibco). Cells were incubated in a humidified atmosphere with 5% CO_2_ at 37 °C. Differentiation of THP-1 monocytes to macrophages was induced by 20 ng/mL of phorbol 12-myristate 13-acetate (PMA) (Sigma) for 48 h in a humidified incubator. Then, cells were washed three times with RPMI 1640 and maintained in complete media for 24 h. To induce inflammation, RAW264.7 and differentiated THP-1 cells were stimulated with lipopolysaccharide (LPS) at 1 μg/mL. Then, cells were incubated without or with different concentrations of CC5 and CC8 for 24 h. The supernatants were stored at −20 °C until use.

### 4.2. CC5 and CC8 Disintegrin Purification

*Cerastes cerastes* snake venom was collected from the Pasteur Institute’s Serpentarium (Tunis, Tunisia) and stored at –20 °C. Crude venom (300 mg) was dissolved in 0.2 M ammonium acetate, pH 6.8, and fractionated by a Sephadex G-75 (Pharmacia, Uppsala Sweden) column. Fraction II, which contains medium-molecular-weight proteins (<30 kDa), was lyophilized subjected to a reverse phase C8 column (250 × 4.6 mm^2^, 5 μm; Beckman; Fullerton, CA, USA) eluted with a linear acetonitrile gradient 10–65% over 45 min at a flow rate of 0.8 mL/min. The homogeneity of CC5 and CC8 were assessed by a second step of high-performance liquid chromatography (HPLC) on a C18 column (250 × 4.6 mm^2^, 5 μm; Beckman) as previously described [21].

### 4.3. Cell Viability Assays

#### 4.3.1. MTT Assay 

Cell viability was assessed by MTT (3-(4, 5-dimethylthiazol-2-yl)-2, 5-diphenyltetrazolium bromide) assay as previously described by Mosmann T [42]. RAW264.7 and differentiated THP-1 were seeded at an appropriate density (5 × 10^3^ cells per well) in a 96-well cell culture plate. Cell number consists of visually counted cells using counting chamber Malassez under an inverted Leica microscope at 10-fold magnification. Then, cells were treated with different concentrations of CC5 and CC8 (0–200 nM). The control cells were not treated with the proteins. After 24 h, 500 µg/mL of MTT were added to the cell medium for 3 h. Then, 100 µL of DMSO was added to dissolve purple precipitate from formazan crystal production, and absorbance was measured at 560 nm.

#### 4.3.2. Trypan Blue Exclusion Assay 

Trypan blue assay was carried as described in [43]. Briefly, the cells were seeded at a density of 0.5 × 10^5^ cells/mL in complete medium. Cells were treated with 200 nM of CC5 and CC8. As MTT assay, the control cells were not treated with the proteins. After 24 h, cells were trypsinized, washed, and resuspended in PBS containing 0.4% trypan blue. The number of viable/dead cells were counted using Counting chamber Malassez as per standard protocol. 

### 4.4. Adhesion Assay

Adhesion assay was performed as previously described by Kadi et al. [44]. First, RAW264.7 and differentiated THP-1 cells were stimulated with LPS (1 μg/mL) for 30 min at 37 °C. Then, cells were incubated in the absence or presence of various concentrations of CC5 and CC8. After 30 min of incubation at 37 °C, cells were seeded at a density of 5 × 10^4^/well in a 96-well plate coated with 30 µg/mL for collagen I (Coll I), vitronectin (Vn), and poly-L-lysine (PLL) and 50 µg/mL for fibrinogen (Fg) during 1 h at 37 °C. Adherent cells were fixed and stained with 0.1% crystal violet. The absorbance was recorded at 600 nm.

### 4.5. Measurement of Cytokine Production in Culture Supernatants 

RAW264.7 and THP-1 cells were pretreated with CC5 and CC8 at different doses (25 and 50 nM), then stimulated with LPS at 1 μg/mL, and the supernatants were collected after 24 h. The quantification of secreted cytokines (IL-6, IL-8, IL-10, and TNF-α) was performed by enzyme-linked immunosorbent assay using the BD OptEIA™ ELISA Kit (BD Biosciences, CA, USA) and mouse ELISA kit (Abcam, Cambridge, UK) according to the manufacturer’s instructions. Cytokine levels were normalized to the cell number. 

### 4.6. Western Blot Analysis

RAW264.7 and differentiated THP-1 macrophages were seeded in 6-well plates. Cells were stimulated with LPS (1 μg/mL) for 30 min and then treated with CC5 and CC8 (25–50 nM) at 37 °C for 24 h. Cell lysates were analyzed by Western blotting. Protein samples were loaded (20 µg/lane), separated by SDS-polyacrylamide gel electrophoresis, and transferred on PVDF (polyvinylidene difluoride, Thermo Fisher Scientific, Waltham, MA, USA) membrane. Then, membranes were incubated in blocking solution (5% non-fat milk) for 1 h and incubated with primary antibodies, including anti-phospho- 65 NF-κB (Serine 536), rabbit monoclonal anti-phospho-Akt (Serine 473), rabbit monoclonal anti-phospho-p38 MAPK (Thr180/Tyr182), anti-phospho-44/42 MAPK (Thr202/Tyr204), anti-integrin αv, anti-integrin β3, and rabbit monoclonal β-actin antibodies. Immunoblots were determined by chemiluminescence HRP-conjugated secondary antibodies, purchased from Cell Signaling Technology (Danvers, MA, USA).

### 4.7. Measurement of Intracellular ROS Generation

Superoxide and hydroxyl radical production in RAW264.7 and differentiated THP1 cells was measured using the Cellular Reactive Oxygen Species Detection Assay Kit (ab186027, Abcam, Cambridge, UK), following the manufacturer’s instructions. Briefly, RAW264.7 and THP-1 macrophages were seeded at a density of 5 × 10^4^ per well in a 96-well plate and stimulated by LPS (1 μg/mL). Subsequently, cells were treated with CC5 and CC8 (25 nM–50 nM) for 24 h. Then, cells were stained with ROS red stock solution for 2 h at room temperature. ROS generation was detected using a fluorescence microplate reader (PerkinElmer, Waltham, MA, USA) at wavelengths of 520 nm for excitation and 605 nm for emission. 

### 4.8. Carrageenan-Induced Paw Edema 

The anti-inflammatory activity of carrageenan-induced paw edema was determined according to the method described by Winter et al. [45]. Young adult male rats of 125–165 g body weight were maintained in air-conditioned quarters with water and food. For the experiment, rats were randomly allocated to five groups of three rats each: the control group received 2.5 mL/kg of physiological solution 0.9% NaCl (used to solubilize the different drugs; the positive control group received 15 mg/kg of carrageenan in 100 µL of 0.9% NaCl); the standard group received 15 mg/kg of carrageenan and 1 mg/kg of dexamethasone; and the test group received 15 mg/kg of carrageenan and 5 µg of CC5 and CC8. The drugs were administered into the left hind paw. Edema was followed by measuring changes in paw volume using a sliding caliper at various times (0, 1, 2, 3, and 4 h). The increase in paw volume was considered as an index of inflammation intensity.

### 4.9. Statistical Analysis 

Data are representative of three independent experiments expressed as mean ± standard deviation (SD). The statistical significance of differential findings between experimental and control groups was determined by Student’s t-test using Graph Pad Prism 4 software. The *p*-values of <0.05 (*), <0.01 (**) and <0.001 (***) were considered to be statistically significant.

## 5. Conclusions

CC5 and CC8 are the first two highly homologous dimeric disintegrins from *Cerastes Cerastes* snake venom that display anti-inflammatory effects both in vitro and in vivo. Indeed, targeting pro-inflammatory signaling pathways is promising for drug conception to treat deadly inflammatory diseases. Thus, synthetic peptidomimetics and either cyclic or linear RGD-derived motifs are a promising source to design integrin antagonists that may be potentially very useful for preclinical studies to further transfer to clinic applications.

## Figures and Tables

**Figure 1 ijms-24-12427-f001:**
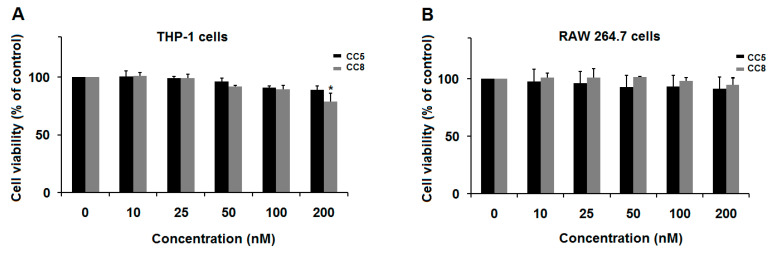
Effect of CC5 and CC8 on cell viability. (**A**) Effect on Human THP-1-derived macrophage viability. (**B**) Effect on RAW264.7 cell viability. Cells were seeded (5 × 10^3^/well) and cultured in presence of CC5 (black bars) or CC8 (grey bars) at different concentrations (10–200 nM) of CC5 and CC8, for 24 h. Non-treated cells were used as negative control. Cell viability was evaluated using MTT assay. Data are representative of three independent experiments performed in triplicate, expressed as mean (±SD). *p* < 0.05 was considered statistically significant and is indicated with asterisks over the value (* *p* < 0.05).

**Figure 2 ijms-24-12427-f002:**
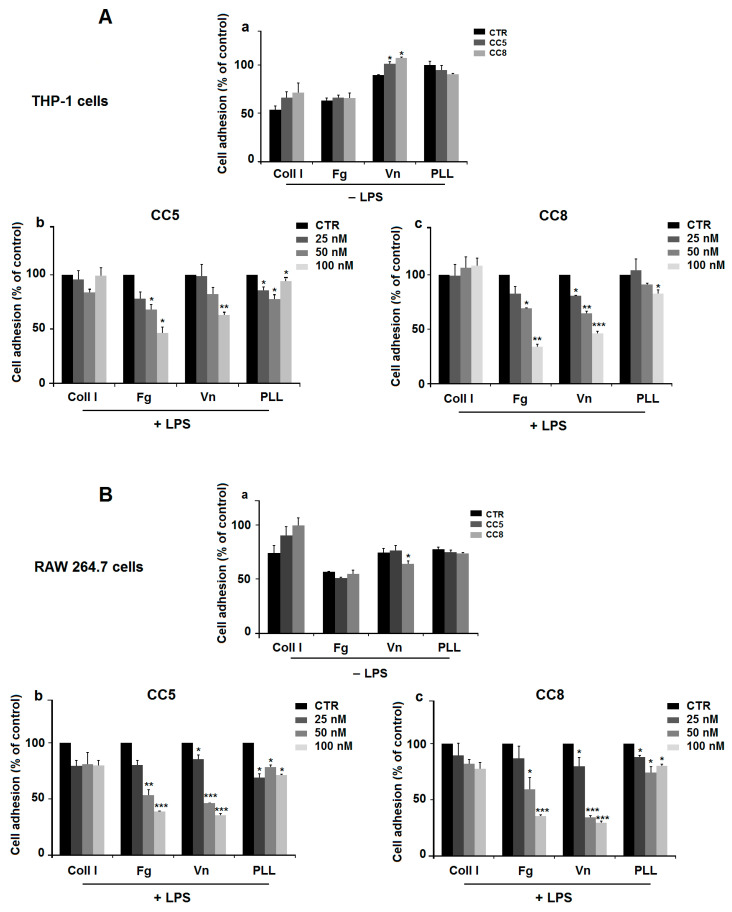
CC5 and CC8 reduce cell adhesion using cell adhesion assay. (**A**) Effect on THP-1-derived macrophage. (**a**) Cells were incubated without (black bars), or with 100 nM CC5 (dark grey bars), or with 100 nM CC8 (light grey bars) for 30 min. Cells were then allowed to attach on fibrinogen (Fg), Collagen I (Coll I), vitronectin (Vn), or poly-L-lysine (PLL). (**b**) Cells were stimulated with LPS (1 µg/mL) and seeded in wells coated with different extracellular matrices and incubated without or with various concentrations of CC5. (**c**) LPS-stimulated THP-1 cells were seeded in wells pretreated with different extracellular matrices and incubated without or with various concentrations of CC8. Non-treated cells were used as negative control. (**B**) Effect on RAW264.7 cells. (**a**) Cells were incubated without (black bars), or with 100 nM CC5 (dark grey bars), or with 100 nM CC8 (light grey bars) for 30 min. Cells were then allowed to attach on Fg, Coll I, Vn, or PLL. (**b**) Cells were stimulated with LPS (1 µg/mL), seeded in wells coated with different extracellular matrices and incubated without or with CC5 at various concentrations. (**c**) LPS-stimulated RAW264.7 cells were seeded in wells pretreated with different extracellular matrices and incubated without or with various concentrations of CC8. Non-treated cells were used as negative control. Data are representative of three independent experiments performed in triplicate, expressed as mean (±SD). *p* < 0.05 was considered statistically significant and is indicated with asterisks over the value (* *p* < 0.05, ** *p* < 0.01, *** *p* < 0.001).

**Figure 3 ijms-24-12427-f003:**
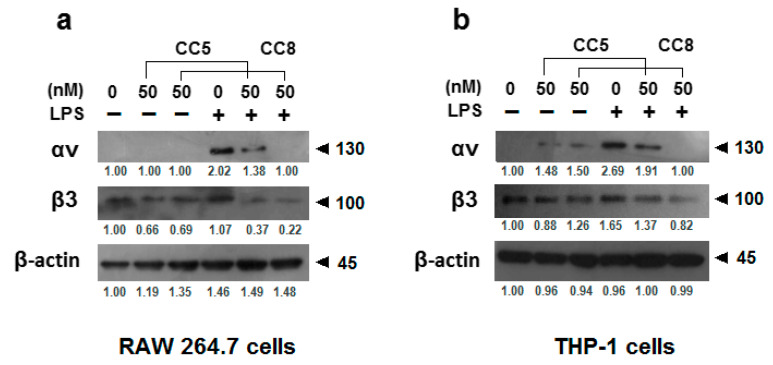
CC5 and CC8 binding with αvβ3 integrin receptor on LPS-stimulated macrophages. (**a**) Human THP-1-derived macrophages were treated or not with LPS (1 μg/mL) in the absence and the presence of CC5 and CC8 (50 nM). (**b**) RAW264.7 cells were treated or not with LPS (1 μg/mL) in the absence and the presence of CC5 and CC8 (50 nM). Cell lysates were analyzed by Western blotting assay, and quantification was performed by using the software program Image J (IJ152). β-actin was used as a loading control. In the presence of LPS, integrin expression in treated cells by protein was compared to untreated cells. All data were statistically significant (*p* < 0.05).

**Figure 4 ijms-24-12427-f004:**
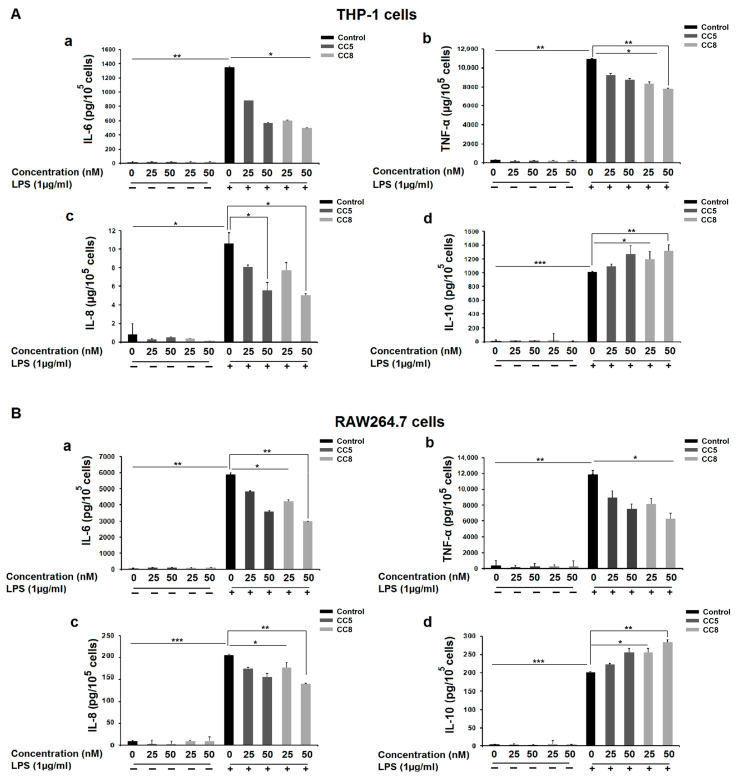
Effect of CC5 and CC8 on inflammatory cytokine release in LPS-stimulated THP-1 and RAW264.7 cells using enzyme-linked immunosorbent assay. (**A**) Human THP-1-derived macrophages were incubated for 24 h in the absence or presence of LPS (1 µg/mL). After treatment with different concentrations of CC5 (dark grey bars) or CC8 (light grey bar), secretions of IL-6 (**a**), TNF-α (**b**), IL-8 (**c**), and IL-10 (**d**) were measured. (**B**) RAW264.7 cells were incubated for 24 h in the absence or presence of LPS (1 µg/mL). After treatment with different concentrations of CC5 (dark grey bars) or CC8 (light grey bars), secretions of IL-6 (**a**), TNF-α (**b**), IL-8 (**c**), and IL-10 (**d**) were measured. The negative controls (black bars) correspond to non-treated cells. Results are reported as the means ± SE of three independent experiments performed in triplicate. *p* < 0.05 was considered statistically significant and is indicated with asterisks over the value (* *p* < 0.05, ** *p* < 0.01, *** *p* < 0.001).

**Figure 5 ijms-24-12427-f005:**
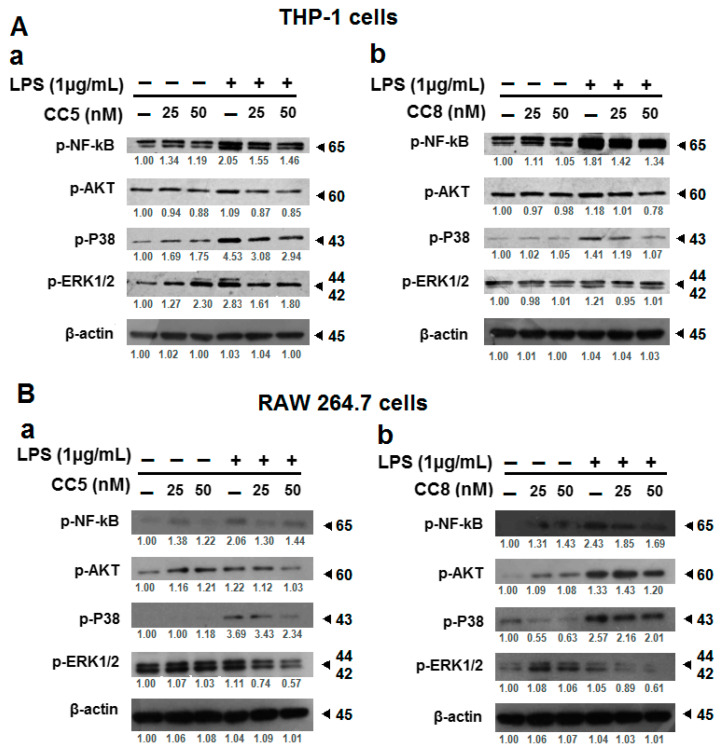
CC5 and CC8 modulate LPS-induced activation of NF-kB, ERK1/2, p38 MAPK, and Akt signaling pathways in LPS-stimulated THP-1 and RAW264.7 cells. (**A**) Cell lysates from human THP-1-derived macrophages untreated or treated with LPS (1 μg/mL) were analyzed by Western blot assay and performed to detect phosphorylated NF-kB, ERK1/2, p38 MAPK, and Akt in the presence or absence of different concentrations of CC5 (**a**) and CC8 (**b**). (**B**) Cell lysates from RAW264.7 untreated or treated with LPS (1 μg/mL) were analyzed by Western blot assay and performed to detect phosphorylated NF-κB, ERK1/2, p38 MAPK, and Akt in the presence or absence of different concentrations of CC5 (**a**) and CC8 (**b**). Protein band quantification was performed using the software program Image J. β-actin was used as a loading control. In LPS-stimulated cells, all data were statistically significant (*p* < 0.05) compared to control (untreated cells with proteins), except p-AKT in THP-1 cells with CC8 (at 25 nM), p-AKT, p-P38 in RAW264.7 cells with CC5 (at 25 nM) and p-AKT, pERK1/2 in RAW264.7 cells with CC8 (at 25 nM).

**Figure 6 ijms-24-12427-f006:**
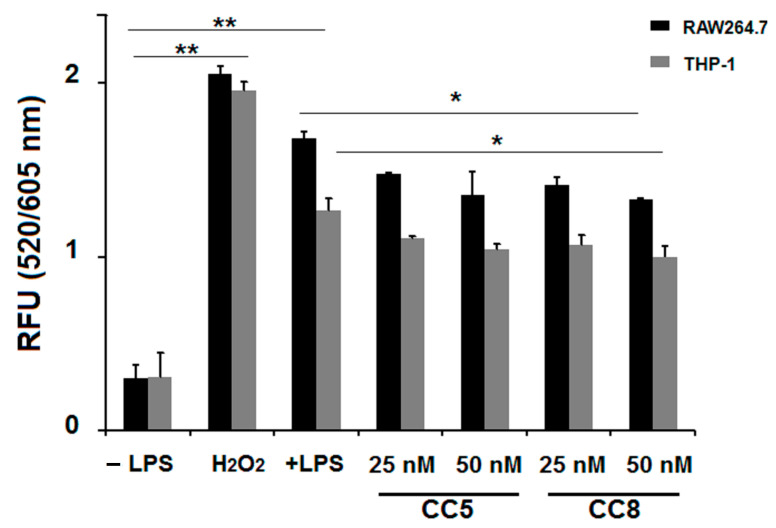
Intracellular ROS production decrease by CC5 and CC8 on LPS-stimulated THP-1 and RAW264.7 cells by measurement with intracellular ROS generation assay. RAW264.7 cells (black bars) and THP-1 cells (grey bars) were stimulated or not with LPS (1 µg/mL) and treated with different concentrations of CC5 and CC8. Untreated cells were used as negative control. H_2_O_2_ (1 mM) was used as positive control. Data are the mean of two independent experiments. *p* < 0.05 was considered statistically significant and is indicated with asterisks over the value (* *p* < 0.05, ** *p* < 0.01).

**Figure 7 ijms-24-12427-f007:**
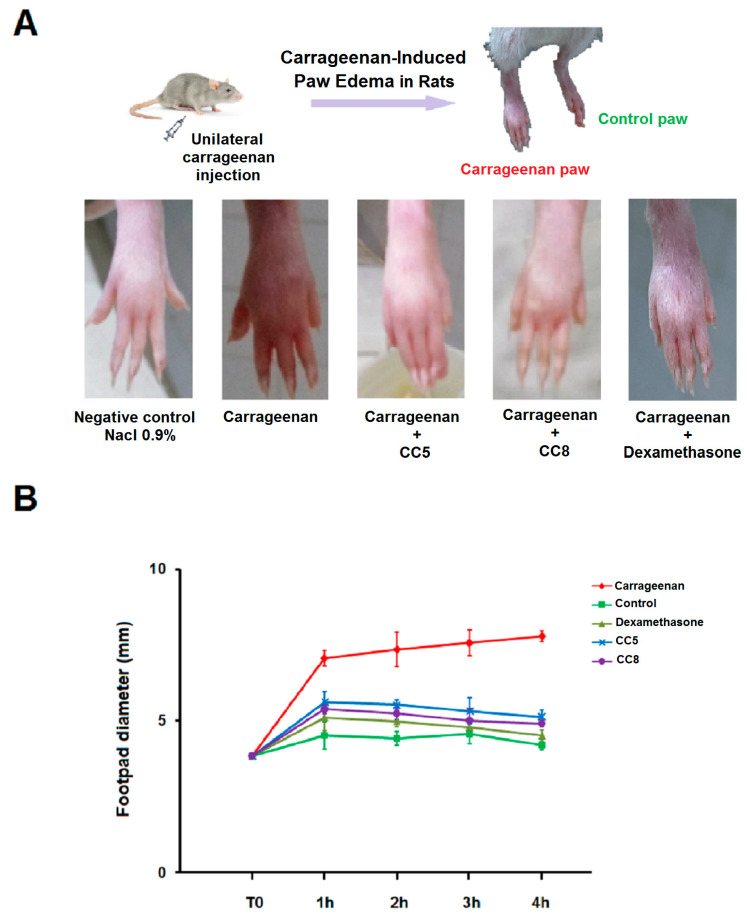
In vivo testing of anti-inflammatory effect of CC5 and CC8 in rats using carrageenan-induced paw edema model. Rats were randomly allocated to five groups of three rats each: Paw edema was induced by injection of carrageenan (15 mg/kg) into rat pads. CC5, CC8 (5 µg), or dexamethasone (1 mg/kg) were administrated by intraperitoneal route. The negative control group received 2.5 mL/kg of physiological solution 0.9% NaCl. (**A**) Macroscopic view of paw edema in different groups for 4 h following carrageenan injection. (**B**) The size of the edema was followed by measuring changes in paw thickness after carrageenan injection using a sliding caliper at various times (0, 1, 2, 3, and 4 h). Data are presented as means ± SD. All data were statistically significant (*p* < 0.05) when compared to the carrageenan group.

## Data Availability

Not applicable.

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
