# Peer review of "CC5 and CC8, Two Disintegrin Isoforms from *Cerastes cerastes* Snake Venom Decreased Inflammation Response In Vitro and In Vivo"

_ijms, 2023, doi:10.3390/ijms241512427_

Round 1
Reviewer 1 Report
This is an interesting study that provides some new perspectives on the study of inflammation involving the innate immune system.
Comments:
1. It is recommended that the abstract be structured.
2. What model was the in vivo study performed on? In the abstract (line 29) and Figure 7, it says there was a mice model, but in section 2.5 (e.g., lines 193, 195) and section 5.7. (line 371) it says there was a rat model.
3. Given that control of immune markers in the mice/rats model of inflammation has not been performed, it is recommended that future research perspectives be described in more detail. In the abstract, the authors point to a potent anti-inflammatory effect on carrageenan-induced edema in the mice model, but immune markers and overall condition of the mice were not evaluated. Perhaps this should be described in the limitation of the study.
4. It is recommended to check the reference list again. For example, reference 29 on line 230. This reference should support information about αvβ3 integrin for some biological processes and diseases, but it is actually a reference to 4-[18F]Fluorobenzoyl-NAVPNLRGDLQVLAQKVART in the Molecular Imaging and Contrast Agent Database (MICAD). In addition, most of the literature sources are significantly older than 5 years.
5. I found no information in the materials and methods about where CC5 and CC8 were obtained.
Author Response
Please see the attachement

Reviewer 2 Report
This is a clear well-crafted study. This is not my primary field of study, but I had no problem following the logic and course of the research you described. This manuscript will make a nice contribution to the field. I recomandate to publish this research.
Author Response
Please see the attachement

Reviewer 3 Report
The authors Morjen et al explored here a new role of disintegrins CC5 and CC8 in inflammation cell response. This question was addressed both in vitro using THP-1 and RAW264.7 macrophages models, and in vivo with a carrageenan-induced edema assay performed in mice model. Through numerous data, the authors highlighted the anti-inflammatory effect of these molecules, whose very probable mechanism seems to involve the alphaV-beta3 integrin/NFkB/MAPK-Akt pathway. Overall, the manuscript is well written and the relevant results are rather well presented. A few minor comments remain, which I'm sure the authors will be able to address, and this work may be published in the journal IJMS.
Major comments:
- In the section Abstract: please review to better explain the objectives of the study.
- In the section Introduction: please, rewrite a few passages to better understand where the authors are going with the subject of their work; another example of improvement, the sentence "The "classical" disintegrins [...] possess an R/K/M/W/VGD [...] sequences..." can be better integrated by "XVD with X being Arg, Lys, Met or Trp residues"; also review the end of the paragraph setting out more precisely the main results and conclusions of their study.
- In the section Results:
- Lines 77-82: please, rewrite the passage for the novice to better introduce the objectives of this part and the method used (MTT test); in addition, it would be useful to confirm the results with another viability test (trypan blue, purple crystal, etc.)
- Lines 124-127 (Fig. 3): I agree with the observations clearly showing a greater effect of CC8 than CC5 used at 50 nM on alphaVbeta3 expression in model cells; on the other hand, there is no difference in the adhesion of stimulated cells in the presence of CC5 or CC8 (fig 2): please, moderate the remarks and conclusions of this part of the text
- Lines 135-144: please, recall the pro- (for IL6, TNFalpha and IL8) or anti- (for IL10) inflammatory effects of these interleukins for the novice
- Lines 166-168: please, bring conclusions to the part of results
- Subsections 2.4 and 2.5: idem please conclusions
- In the section Discussion:
The discussion is generally well written and the results obtained well placed in the scientific context. Be careful, the authors use another unit of measurement (in μg/mL) than in the text (in nM) to compare the effects of CC5 and CC8 to other molecules already described in the literature (trimucrin, etc.), which makes it difficult to make the comparison relevant for the novice who does not know the MM of these molecules!
Minor comments:
- please, check the legend of several figures (e.g.: figures 2 and 3) in which it is not specified neither the type of test nor the conditions compared in the studies
- Legend of Fig. 7: please, specify the number of legs or animals taken into consideration for the quantification
- Please, review also the text of the manuscript, many typos are present.

Reviewer 4 Report
Review of the article: CC5 and CC8, two disintegrin isoforms from Cerastes cerastes snake
venom decreased inflammation response in vitro and in vivo
Manuscript ID: ijms-2497082
In this study, the authors investigated the anti-inflammatory effect of CC5 and CC8 in LPS-stimulated cells in vitro and their underlying mechanisms, as well as evaluated the effect in vivo by carrageenan-induced edema in mice model. The information in this study would be beneficial to design integrin antagonists that may be potentially useful for preclinical studies to further transfer to clinic applications. I have few comments that the authors should notice and reply, as bellows.
1. In line 87, “as control” should be revised.
2. In line 94, the authors mentioned that CC5 and CC8 did not affect the adherence of non-stimulated cells seeded on ECM. But the data shown in Figure 2A(a) and 2B(a) revealed that the adherence to vitronectin (Vn) with CC5 or CC8 was significantly different to control (CTR).
3. In lines 111 and 117, “Cl” should be revised.
4. In Figure 3(a), the quantity of beta-actin seems very different among different lanes (i.e., the thickness of the 5th band seems more than 1.5 fold of the first one), which is inconsistent to the very similar numerals (1.00-1.07) below corresponding bandings.
5. In line 178, “All data were statistically significant (p < 0.05)” should be revised.
6. At line 185, “…17.6% and 19.3% respectively for CC5, 21% and 20.8% respectively for CC8” is ambiguous. Which cell line do these values refer to, respectively?
7. At lines 231-234, how do you certify the concentration of CC5 and CC8 with maximal inhibitory effect is ≃ 1.4 μg/ml from the very limited concentration groups of CC5 and CC8 in your study?
8. In Materials and Methods, 1) the methods to purify and certify CC5 and CC8 are not presented. 2) the method to quantify the cell number is not presented.
9. In line 375, did the group receiving dexamethasone also receive carrageenan?
10. In References, the format of texts in several places need to be revised, such as those at lines 411-412, 467-468, 484, 488-489, and 508.
Round 2
Reviewer 1 Report
The authors made changes that improved the quality of the article. It would be useful to add a section on the limitations of the study, where, for example, to indicate the lack of control of immune markers on the carrageenan-induced paw edema model.
Reviewer 4 Report
The authors have resolved all my questions.